# Mode-Aware GAN: Continual Adaptation for Conditional Image Generation

## Abstract

Continuously learning new modes in generative models while preserving previously learned ones is a significant challenge, particularly with limited training samples. Here, we propose a Mode Affinity Score tailored for continual learning within conditional generative adversarial networks. This score, derived from the discriminators, measures the similarity between generative tasks. By leveraging this score, new modes can be seamlessly integrated into the model through an interpolation process among the closest learned modes, guided by the computed affinity scores. This approach enhances generation performance and mitigates the risk of catastrophic forgetting. Extensive experiments demonstrate the efficacy of our method compared to existing techniques, even when using significantly fewer training samples.

## 1 Introduction

Generative artificial intelligence has made significant progress in recent years, and we have seen remarkable applications, such as ChatGPT (OpenAI, 2021), and DALL-E (Vaswani et al., 2021). Nevertheless, most of these methods Wang et al. (2018); Zhai et al. (2019); Seff et al. (2017) lack the ability to learn continuously, which remains a challenging problem in developing generative AI models that can match human's continuous learning capabilities. This challenge is particularly difficult when the target data is limited (Varshney et al., 2021). In this scenario, the objective is to generate target images with a few training samples while retaining previously acquired knowledge from earlier tasks. Most of the continual learning methods (Zhai et al., 2019; Seff et al., 2017) focus on preventing the models from forgetting the existing tasks through multiple enforced learning restrictions that often lead to poor performance on new tasks.

A solution for efficiently learning new tasks is to identify and leverage the relevant knowledge from previously learned tasks. Based on this principle, various knowledge transfer approaches have been introduced, resulting in significant breakthroughs in many applications, including natural language processing (Devlin et al., 2018), and computer vision (Elaraby et al., 2022; Azizi et al., 2021). These techniques enable models to leverage past experiences, such as trained weights, and hyper-parameters to improve the performance of the new task, emulating how humans learn and adapt to new challenges (e.g., riding motorcycles is less challenging for people who know how to ride bicycles). It is also essential to identify the most relevant task for knowledge transfer when dealing with multiple learned tasks. Irrelevant knowledge can be harmful when learning new tasks (Le et al., 2022b; Standley et al., 2020b), resulting in flawed conclusions. For example, incorrectly classifying dolphins as fishes instead of mammals could lead to misconceptions about their reproduction.

Here, we propose a *Discriminator-based Mode Affinity Score* (dMAS) to evaluate the similarity between generative tasks and introduce a new continual learning framework for Generative Adversarial Network (Mirza & Osindero, 2014), called Mode-aware GAN. By identifying and utilizing suitable information from previously learned tasks, our approach allows for seamless and efficient integration of new tasks into the continual learning models. Our model first evaluates the similarity between the existing modes and the target data using dMAS. It enables the identification of the closest or the most relevant modes whose knowledge can be leveraged for quick adaptation of the target data while preserving the knowledge of the existing modes. To this end, we add a new mode to the generative model to represent the target task. This mode is assigned

an embedding label derived from the embeddings of the closest modes and the computed dMAS values between the closest modes and the target. We incorporate generative replay (Chenshen et al., 2018) to further mitigate catastrophic forgetting.

Extensive experiments are conducted on the MNIST (LeCun et al., 2010), CIFAR-10 (Krizhevsky et al., 2009), CIFAR-100 (Krizhevsky et al., 2009), ImageNet (Russakovsky et al., 2015), Oxford Flower (Nilsback & Zisserman, 2008), and CelebA (Liu et al., 2015) datasets to validate the efficacy of our proposed framework. We first empirically demonstrate the stability and robustness of dMAS, showing that it remains invariant across different model settings. Utilizing this affinity, the proposed framework effectively utilizes knowledge from the learned modes for learning the new tasks, significantly reducing the required data samples in both transfer learning and continual learning scenarios. We achieve competitive results compared with baselines and the state-of-the-art approaches, including sequential fine-tuning (Wang et al., 2018), multi-task learning (Standley et al., 2020b), EWC-GAN (Seff et al., 2017), Lifelong-GAN (Zhai et al., 2019), and CAM-GAN (Varshney et al., 2021). Notably, our framework demonstrates a significant performance improvement for the target task, with only a small performance loss for previously learned tasks. Moreover, the average performance consistently increases across each learning iteration.

## 2 Related Works

Continual learning involves the problem of learning a new task while avoiding catastrophic forgetting (Kirkpatrick et al., 2017; McCloskey & Cohen, 1989; Carpenter & Grossberg, 1987). It has been extensively studied in image classification (Verma et al., 2021; Zenke et al., 2017; Wu et al., 2018; Singh et al., 2020; Rajasegaran et al., 2020). In image generation, previous works have addressed continual learning for a small number of tasks or modes in GANs (Mirza & Osindero, 2014). These approaches, such as memory replay (Wu et al., 2018), have been proposed to prevent catastrophic forgetting (Zhai et al., 2019; Cong et al., 2020; Rios & Itti, 2018). However, as the number of modes increases, network expansion (Yoon et al., 2017; Xu & Zhu, 2018; Zhai et al., 2020; Mallya & Lazebnik, 2018; Masana et al., 2020; Rajasegaran et al., 2019) becomes necessary to efficiently learn new modes while retaining previously learned ones. Nevertheless, the excessive increase in the number of parameters remains a major concern.

The concept of task similarity has been widely investigated in transfer learning, which assumes that similar tasks share common knowledge that can be transferred from one to another. However, existing approaches in transfer learning (Silver & Bennett, 2008; Finn et al., 2016; Mihalkova et al., 2007; Niculescu-Mizil & Caruana, 2007; Luo et al., 2017; Razavian et al., 2014; Pan & Yang, 2010; Chen et al., 2018) mostly focus on sharing the model weights from the learned tasks to the new task without explicitly identifying the closest tasks. In recent years, several works (Le et al., 2022b; Zamir et al., 2018; Pal & Balasubramanian, 2019; Dwivedi & Roig., 2019; Achille et al., 2019; Wang et al., 2019; Standley et al., 2020a) have investigated the relationship between image classification tasks and applied relevant knowledge to improve overall performance. However, for the image generation tasks, the common approaches to quantify the similarity between tasks or modes are using common image evaluation metrics, such as Fréchet Inception Distance (FID) (Heusel et al., 2017) and Inception Score (IS) (Salimans et al., 2016). While these metrics can provide meaningful similarity measures between two distributions of images, they do not capture the state of the model and therefore may not be suitable for transfer learning and continual learning. For example, a model trained to generate images for one task may not be useful for another task because this model is not well-trained, even if the images for both tasks are visually similar.

In the field of continual learning for image generation (Wang et al., 2018; Varshney et al., 2021; Zhai et al., 2019; Seff et al., 2017), mode-affinity has not been explicitly considered. Although some prior works (Zhai et al., 2019; Seff et al., 2017) have explored fine-tuning GAN models (Arjovsky et al., 2017; Zhu et al., 2017) with regularization techniques, such as Elastic Weight Consolidation (Kirkpatrick et al., 2017), or the Knowledge Distillation (Hinton et al., 2015), they did not focus on measuring mode similarity or selecting the closest modes for knowledge transfer. Other approaches use different assumptions such as global parameters for all modes and individual parameters for particular modes (Varshney et al., 2021). Their proposed task distances also require a well-trained target generator, making them unsuitable for real-world continual learning scenarios.

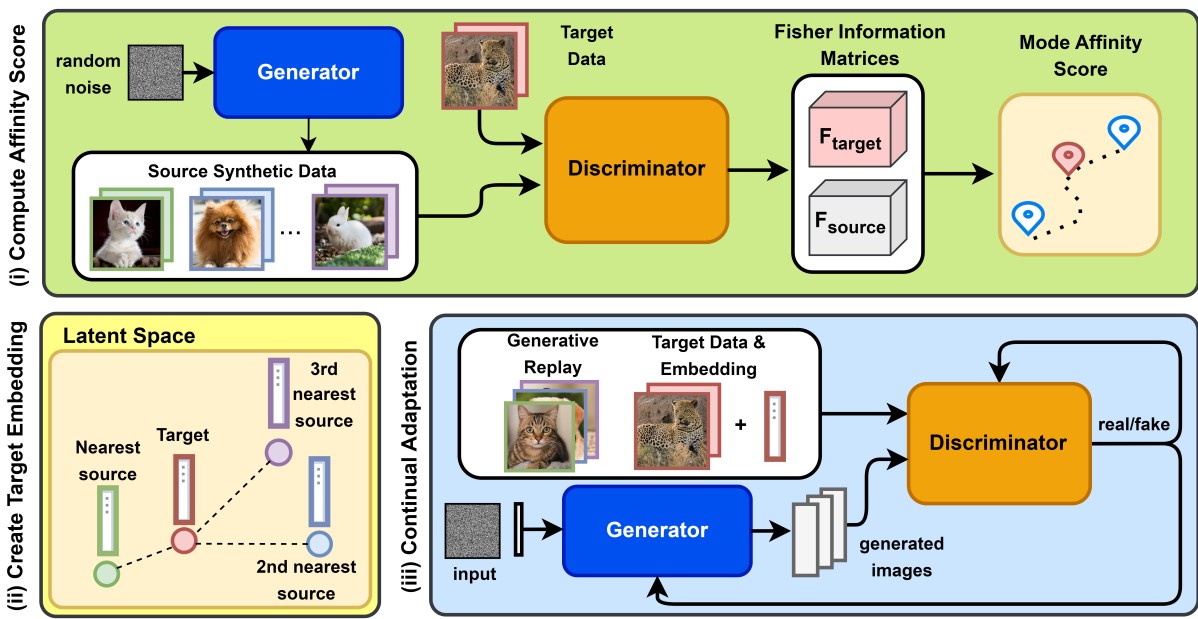

Figure 1: The overview of mode-aware continual learning framework for generative adversarial networks: (i) Utilizing the pre-trained GAN model to compute the mode affinity scores from source modes to the target, (ii) Constructing a set of closest modes based on the computed distance and creating the target label embedding using the embeddings from the closest modes, (iii) Fine-tuning the model using the target data and the newly-generated embedding.

## 3 Mode Affinity Score

Consider a generative adversarial network that is trained on a set $S$ of source generative tasks, where each task represents a distinct class of data. The model consists of two key components: the generator $\mathcal{G}$ and the discriminator $\mathcal{D}$. Each source generative task $a \in S$, which is characterized by data $X_a$ and its labels $y_a$, corresponds to a specific *mode* in the well-trained generator $\mathcal{G}$. Let $X_b$ denote the target data.

In this paper, we propose a new task-affinity measure, called Discriminator-based Mode Affinity Score (dMAS), to showcase the complexity involved in transferring knowledge between different modes in GAN. This measure is based on the Fisher Information matrices and is approximated by computing the expectation of Hessian matrices from the discriminator's loss function. Particularly, we input the source synthetic data $X_a$ into the discriminator $\mathcal{D}$ to compute the corresponding loss. By taking the second-order derivative of the discriminator's loss with respect to the input, we obtain the source Hessian matrix. Similarly, we repeat this process using the target data $X_b$ as input to the discriminator, resulting in the target Hessian matrix. These matrices offer valuable insights into the significance of the model's parameters concerning the desired data distribution. In other words, it indicates whether the input samples are closely related to the training data. The dMAS is defined as the Fréchet distance between these Hessian matrices.

**Definition 1** (Discriminator-based Mode Affinity Score)**.** *Consider a well-trained conditional GAN with discriminator $\mathcal{D}$ and the generator $\mathcal{G}$ that has $S$ learned modes. For the source mode $a \in S$, let $X_a$ denote the real data, and $\tilde{X}_a$ be the generated data from mode $a$ of the generator $\mathcal{G}$. Given $X_b$ is the target real data, $H_a, H_b$ denote the expectation of the Hessian matrices derived from the loss function of the discriminator $\mathcal{D}$ using $\{X_a, \tilde{X}_a\}$ and $\{X_b, \tilde{X}_a\}$, respectively. The distance from the source mode $a$ to the target $b$ is defined to be:*

$$s[a, b] := \frac{1}{\sqrt{2}} \textbf{\textit{trace}}\left( H_a + H_b - 2H_a^{1/2}H_b^{1/2} \right)^{1/2}. \tag{1}$$

To simplify Equation (1), we approximate the Hessian matrices with their normalized diagonals as computing the full Hessian matrices in the large parameter space of neural networks can be computationally expensive.

Hence, dMAS can be expressed as follows:

$$s[a, b] = \frac{1}{\sqrt{2}} \left\| H_a^{1/2} - H_b^{1/2} \right\|_F \tag{2}$$

The procedure to compute dMAS is outlined in function `dMAS()` in Algorithm 1. Our metric spans a range from 0 to 1, where 0 signifies a perfect match, while 1 indicates complete dissimilarity. It is important to note that dMAS exhibits an asymmetric nature, reflecting the inherent ease of knowledge transfer from a complex model to a simpler one, as opposed to the reverse process.

In contrast to the statistical biases observed in metrics such as IS (Salimans et al., 2016) and FID (Heusel et al., 2017; Chong & Forsyth, 2020), dMAS is purposefully crafted to cater to our specific scenario of interest. It takes into account the state of the GAN model, encompassing both the discriminator and the generator. This sets it apart from FID, which uses the GoogleNet Inception model to measure the Wasserstein distance to the ground truth distribution. Consequently, it falls short in evaluating the quality of generators and discriminators. Instead of assessing the similarity between Gaussian-like distributions, our proposed dMAS quantifies the Fisher Information distance between between the model weights. Thus, it accurately reflects the current states of the source models. Furthermore, FID has exhibited occasional inconsistency with human judgment, leading to suboptimal knowledge transfer performance (Liu et al., 2018; Wang et al., 2018). In contrast, our measure aligns more closely with human intuition and consistently demonstrates its reliability. It is important to emphasize that dMAS is not limited to the analysis of image data samples; it can be effectively applied to a wide range of data types, including text and multi-modal datasets.

## 4    Mode-Aware GAN

We introduce the continual learning framework using the mode affinity score for image generation. The goal is to train a continual learning GAN model to learn new modes while avoiding catastrophic forgetting of existing modes. Consider a scenario where each generative task represents a distinct class of data. At time $t$, the model has $S$ modes corresponding to $S$ learned tasks.

Here, we propose a *mode-aware continual learning* framework that allows the model to add a new mode while retaining knowledge from previous modes. We begin by embedding the numeric label of each data sample, using an embedding layer in both the generator $\mathcal{G}$ and the discriminator $\mathcal{D}$ models. We then modify the model to enable it to take a linear combination of label embeddings for the target data. These label embeddings correspond to the most relevant modes, and the weights for these embedding features are associated with the computed dMAS from the related modes to the target. This enables the model to add a new target mode while maintaining all existing modes. Let `emb()` denote the output of the embedding layers in the generator $\mathcal{G}$ and the discriminator $\mathcal{D}$, and $C$ be the set of the relevant source modes, $C = \{i_1^*, i_2^*, \ldots, i_n^*\}$. The computed mode-affinity scores from these modes to the target are denoted as $s_i^*$. Let $\sum_{i=1}^{n} s_i^*$ denote the total distance from all the relevant modes to the target. The label embedding for the target data samples is described as follows:

$$\text{emb}(y_{target}) = \sum_{j \in C} \frac{\sum_{i=1}^{n} s_i^* - s_j}{\sum_{i=1}^{n} s_i^*} \text{emb}(y_{source_j}) \tag{3}$$

In order to inject the target mode into the model without forgetting the existing learned modes, we use the target data with the above label embedding to train the model. Additionally, we utilize generative replay (Robins, 1995; Chenshen et al., 2018) to further mitigate catastrophic forgetting. Particularly, samples generated from relevant source modes are used to fine-tune the model. The overview of the proposed approach is illustrated in Figure 1. During each iteration, training with the target data and generative replay are jointly implemented using an alternative optimization process. The pseudocode of the framework is provided in Algorithm 1. By applying the closest modes' label embeddings to construct the target embedding, we can precisely update part of the model without sacrificing the generation performance of other existing modes. Overall, utilizing knowledge from past experience helps enhance the performance of the model in learning new modes while reducing the amount of the required training data samples. Next, we provide a theoretical analysis of our proposed method.

---

**Algorithm 1:** Mode-Aware Continual Learning for Generative Adversarial Networks

---

**Data:** Source data: $(X_{source}, y_{source})$, Target data: $X_{target}$
**Input:** The generator $\mathcal{G}$ and discriminator $\mathcal{D}$ of GAN
**Output:** Continual learning generator $\mathcal{G}_{\bar{\theta}_g}$ and discriminator $\mathcal{D}_{\bar{\theta}_d}$
**Function** dMAS$(y_a, X_b, \mathcal{G}, \mathcal{D})$:

    Generate data $\tilde{X}_a$ of class label $y_a$ using the generator $\mathcal{G}$, and set $X_a = \tilde{X}_a$
    Compute $H_a$ from the loss of discriminator $\mathcal{D}$ using $\{X_a, \tilde{X}_a\}$
    Compute $H_b$ from the loss of discriminator $\mathcal{D}$ using $\{X_b, \tilde{X}_a\}$
    **return** $s[a,b] = \frac{1}{\sqrt{2}} \left\| H_a^{1/2} - H_b^{1/2} \right\|_F$

**Function** Main:

    Construct S source modes, each from a data class in $y_{source}$     ▷ Pre-train GAN model
    Train $(\mathcal{G}_{\theta_g}, \mathcal{D}_{\theta_d})$ with $X_{source}, y_{source}$
    **for** $i = 1, 2, \ldots, S$ **do**
        $s_i = $ dMAS$(y_{source_i}, X_{target}, \mathcal{G}_{\theta_g}, \mathcal{D}_{\theta_d})$     ▷ Find the closest modes
    **return** closest mode(s): $i^* = \underset{i}{\arg\min}\, s_i$
    Create a set of closest mode(s) $C = \{i_1^*, i_2^*, \ldots, i_n^*\}$
    Generate the target label embedding: emb$(y_{target}) = \sum_{j \in C} \frac{\sum_{i=1}^n s_i^* - s_j}{\sum_{i=1}^n s_i^*}$ emb$(y_{source_j})$
    **while** $\theta$ *not converged* **do**
        Update $\mathcal{G}_{\theta_g}, \mathcal{D}_{\theta_d}$ using $X_{target}$ and emb$(y_{target})$     ▷ Fine-tune for continual learning
        Replay $\mathcal{G}_{\theta_g}, \mathcal{D}_{\theta_d}$ with $X_{source_{i*}}$
    **return** $\mathcal{G}_{\bar{\theta}_g}, \mathcal{D}_{\bar{\theta}_d}$

---

**Theorem 1.** *Let $\theta$ be the model's parameters and $X_a, X_b$ be the source and target data, with the density functions $p_a, p_b$, respectively. Assume the loss functions $L_a(\theta) = \mathbb{E}[l(X_a; \theta)]$ and $L_b(\theta) = \mathbb{E}[l(X_b; \theta)]$ are strictly convex and have distinct global minima. Let $X_n$ be the mixture of $X_a$ and $X_b$, described by $p_n = \alpha p_a + (1-\alpha) p_b$, where $\alpha \in (0,1)$. The corresponding loss function is $L_n(\theta) = \mathbb{E}[l(X_n; \theta)]$. Under these assumptions, it follows that $\theta^* = \arg\min_\theta L_n(\theta)$ satisfies:*

$$L_a(\theta^*) > \min_\theta L_a(\theta) \tag{4}$$

In the above theorem, the introduction of a new mode through mode injection inherently involves a trade-off between the mode-adding ability and the potential performance loss compared to the original model. In essence, when incorporating a new mode, the performance of existing modes cannot be improved. In other words, the main goal of our approach is to minimize the performance loss for the existing modes, while maximizing the performance on the target to boost the overall generative performance. The detailed proof of Theorem 1 is provided in Appendix A.

## 5 Experimental Study

In these experiments, we evaluate the effectiveness of the proposed mode-affinity measure in the continual learning framework, as well as the consistency of the discriminator-based mode affinity score. We consider a scenario where each generative task corresponds to a single data class in the MNIST (LeCun et al., 2010), CIFAR-10 (Krizhevsky et al., 2009), CIFAR-100 (Krizhevsky et al., 2009), ImageNet (Russakovsky et al., 2015), Oxford Flower (Nilsback & Zisserman, 2008), and CelebA (Liu et al., 2015) datasets. Here, we compare the proposed framework with baselines and state-of-the-art approaches, including individual learning (Arjovsky et al., 2017), sequential fine-tuning (Wang et al., 2018), multi-task learning (Standley et al., 2020b), FID-transfer learning (Wang et al., 2018), EWC-GAN (Seff et al., 2017), Lifelong-GAN (Zhai et al., 2019), and CAM-GAN (Varshney et al., 2021). The results show the efficacy of our approach in terms

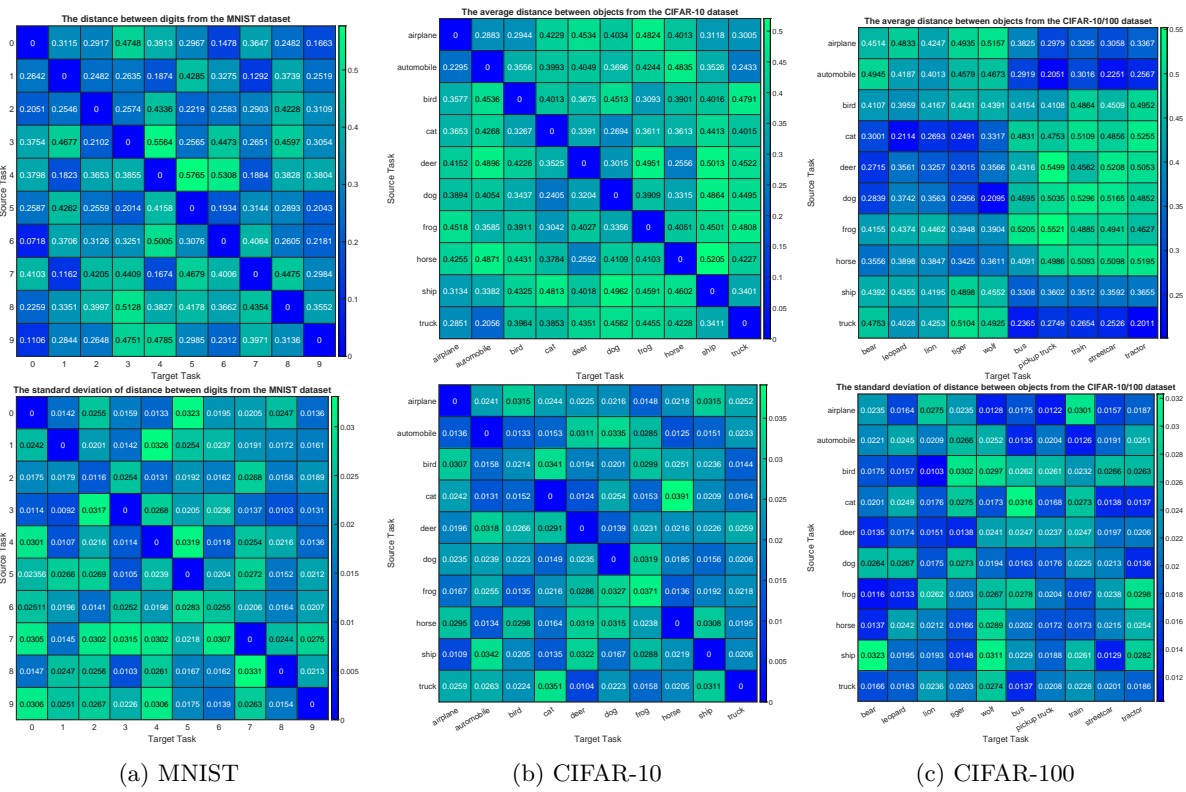

Figure 2: The mean (top) and standard deviation (bottom) of computed mode-affinity scores between data classes of the MNIST, CIFAR-10, CIFAR-100 dataset.

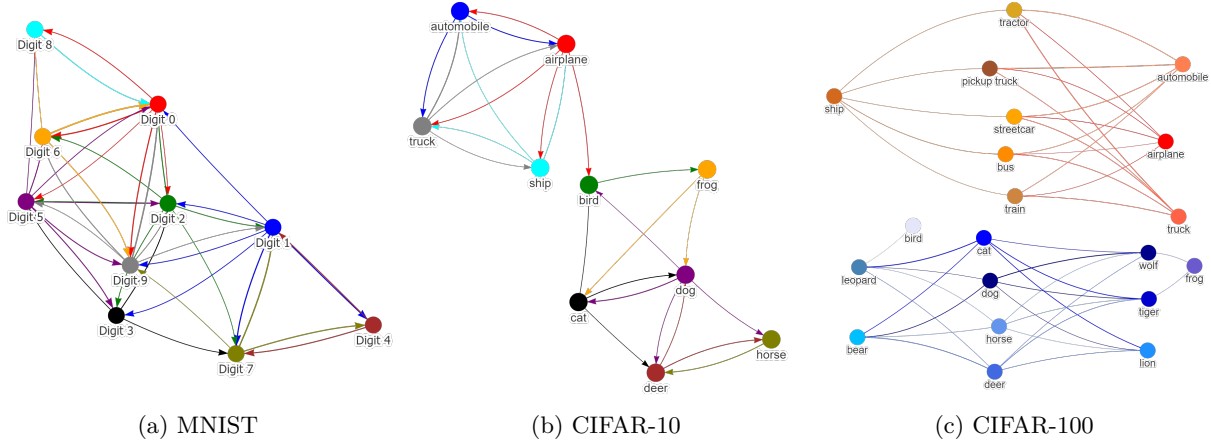

Figure 3: The the atlas plots computed across 10 trial runs for data classes from (a) MNIST, (b) CIFAR-10, and (c) CIFAR-100.

of generative performance and the ability to learn new modes while preserving knowledge of the existing modes.

## 5.1 Mode Affinity Score Consistency

In this experiment, 10 generative tasks are defined based on the MNIST dataset, where each task corresponds to generating a specific digit. The GAN model was trained to generate images from the 9 source tasks while

Table 1: Knowledge transfer performance using mode affinity score against other baselines and FID transfer learning approaches for 10-shot, 20-shot, and 100-shot in CIFAR-100 datasets.

| Approach | 10-shot | 20-shot | 100-shot |
|---|---|---|---|
| Individual Learning (Arjovsky et al., 2017) | 94.82 | 89.01 | 78.47 |
| Sequential Fine-tuning (Zhai et al., 2019) | 88.03 | 79.51 | 67.33 |
| Multi-task Learning (Standley et al., 2020b) | 80.06 | 76.33 | 61.59 |
| FID-Transfer Learning (Wang et al., 2018) | 61.34 | 54.18 | 46.37 |
| **MA-Transfer Learning (ours)** | **57.16** | **50.06** | **41.81** |

considering the remaining task as the target. Here, the generator serves as the representation network for the source data. To evaluate the consistency of the closest modes for each target, we conduct 10 trial runs, in which the source model is initialized randomly. The mean and standard deviation of the mode-affinity scores between each pair of source-target modes are shown in Figure 3 (a) and Figure 2, respectively. In the mean table, the columns denote the mode affinity score from each source mode to the given target. The standard deviation table indicates that the calculated distance is stable, as there are no overlapping fluctuations and the orders of similarity between tasks are preserved across 10 runs. This suggests that the tendency of the closest modes for each target remains consistent regardless of the initialization of the model. Thus, the computed mode affinity score demonstrates consistent results. We provide the atlas plot in Figure 3(a) which highlights the relationship between the digits based on the computed distances. The plot reveals that digits $1, 4, 7$ exhibit a notable similarity, while digits $0, 6, 8$ are closely related.

Similarly, we evaluate the consistency of the mode affinity scores on the CIFAR-10 and CIFAR-100 datasets. For the CIFAR-10 dataset, we define 10 tasks, each corresponding to a specific object. As in the previous experiment, one task is designated as the target task, while the others serve as source tasks for training the model in image generation. The mean and standard deviation of the computed mode affinity scores between CIFAR-10 tasks are shown in Figure 3(b) and Figure 2, respectively. Additionally, Figure 3(b) presents an atlas plot that provides an overview of the relationships between the objects based on the computed mode affinity scores. This plot reveals a strong connection among automobile, truck, ship, and airplane classes, while the remaining classes also show significant resemblance. For the CIFAR-100 dataset, we select 10 image classes (bear, leopard, lion, tiger, wolf, bus, pickup truck, train, streetcar, and tractor) and define 10 target tasks, with each task corresponding to a specific image class. In this experiment, the model is trained on 10 generative tasks in the CIFAR-10 dataset. This model is then used to compute the mode affinity scores from the CIFAR-10 tasks to the CIFAR-100 tasks. Figure 3(c) and Figure 2 respectively display the mean and standard deviation of the computed mode affinity scores between the source and target tasks. The mean table indicates the average distance from each CIFAR-10 source mode to the CIFAR-100 target mode. Notably, the target tasks for generating bear, leopard, lion, tiger, and wolf images are closely related to the cat, deer, and dog groups from CIFAR-10. Specifically, cat images are closely related to leopard, lion, and tiger images. Furthermore, the target tasks for generating bus, pickup truck, streetcar, and tractor images are highly related to the automobile, truck, airplane, and ship group from CIFAR-10. Moreover, Figure 3(c) includes an atlas plot that visually represents the relationships between objects based on the computed distances. This plot reveals strong connections among the vehicle classes and notable closeness among the animal classes.

Additionally, we conducted a knowledge transfer experiment to evaluate the effectiveness of the proposed mode affinity score (dMAS) in transfer learning scenarios using the MNIST, CIFAR-10, and CIFAR-100 datasets. In these experiments, one data class was designated as the target, while the remaining nine classes served as source tasks. Our method first computed the dMAS distance from the target to each source task. After identifying the closest task, we fine-tuned the GAN model using the target data samples labeled from the closest task. This approach allowed the model to update specific parts efficiently, facilitating quicker learning of the target task. The image generation average performance over 10 tasks in CIFAR-100, measured by FID scores, is presented in Table 1. Notably, our utilization of dMAS for knowledge transfer significantly outperformed other methods while using only 10% of the target training samples. Compared to FID-based

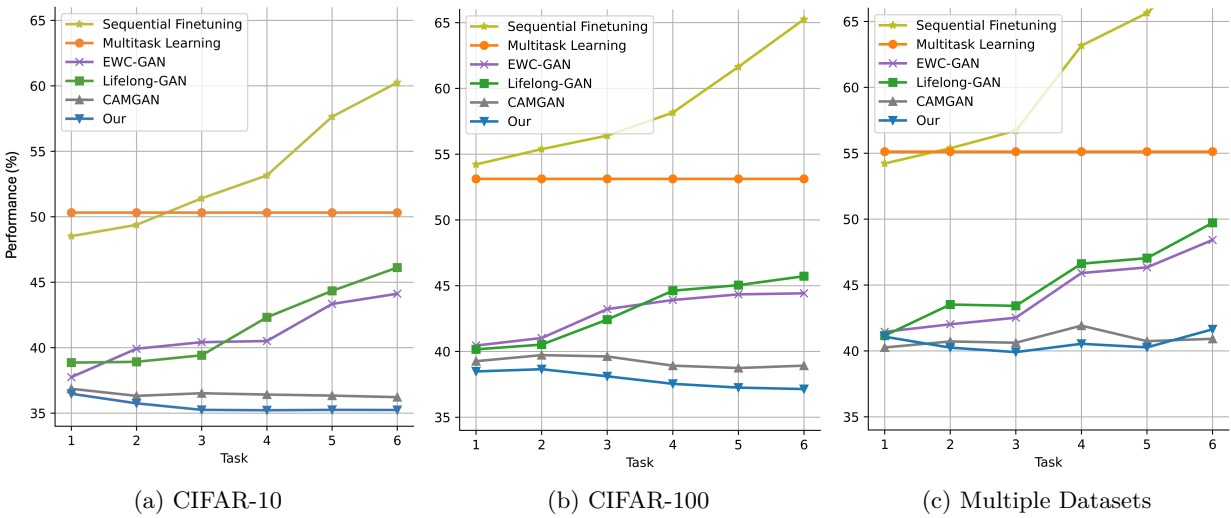

(a) CIFAR-10         (b) CIFAR-100         (c) Multiple Datasets

Figure 4: The continual learning performance over all learned tasks for target tasks from (a) CIFAR-10, (b) CIFAR-100, and (c) the combination of ImageNet, Flower, and CelebA datasets.

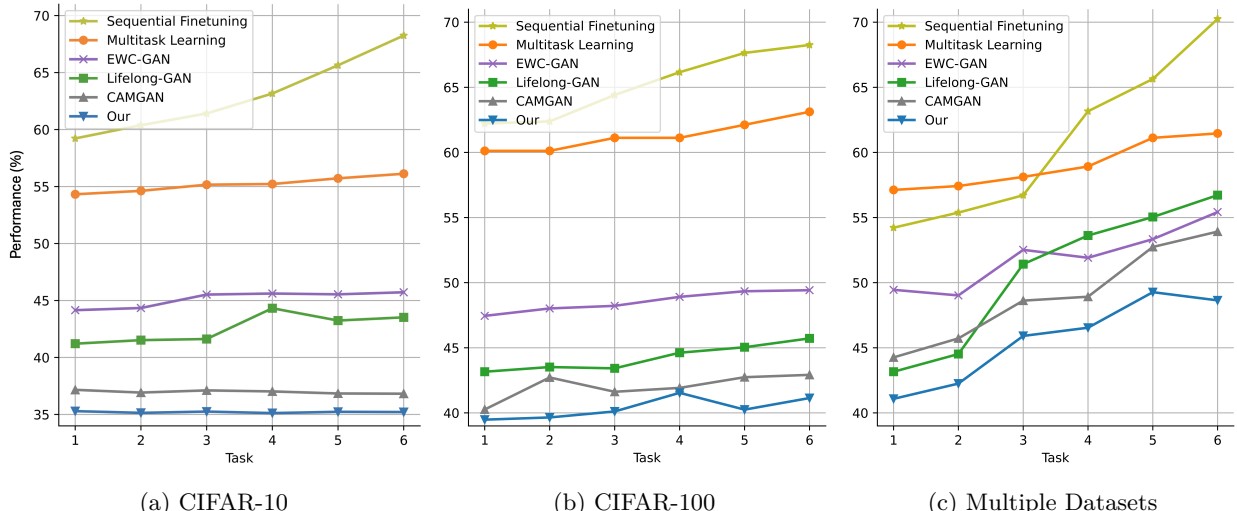

(a) CIFAR-10         (b) CIFAR-100         (c) Multiple Datasets

Figure 5: The continual learning performance of the target tasks from (a) CIFAR-10, (b) CIFAR-100, and (c) the combination of ImageNet, Flower, and CelebA datasets.

transfer learning (Wang et al., 2018), our approach achieved better results in 10-shot, 20-shot, and 100-shot scenarios. Overall, our method considers the state of the models and selects the task closest to the target images, resulting in more effective knowledge transfer. This strategic selection enables the model to leverage pre-existing knowledge more efficiently, leading to improved performance in generative tasks.

## 5.2 Continual Learning Performance

In this experiment, we evaluate the performance of our continual adaptation approach for generative tasks across the CIFAR-10, CIFAR-100, ImageNet, Flower, and CelebA datasets. First, we consider 6 generative tasks from CIFAR-10 as the targets for continual learning. The GAN model is initially pretrained on source tasks, which consist of the remaining tasks from the CIFAR-10 dataset. Subsequently, this model is continually adapted to the target tasks using the mode-aware continual learning framework. In our approach, we select the top-2 closest modes to each target and leverage their knowledge for quick adaptation to the

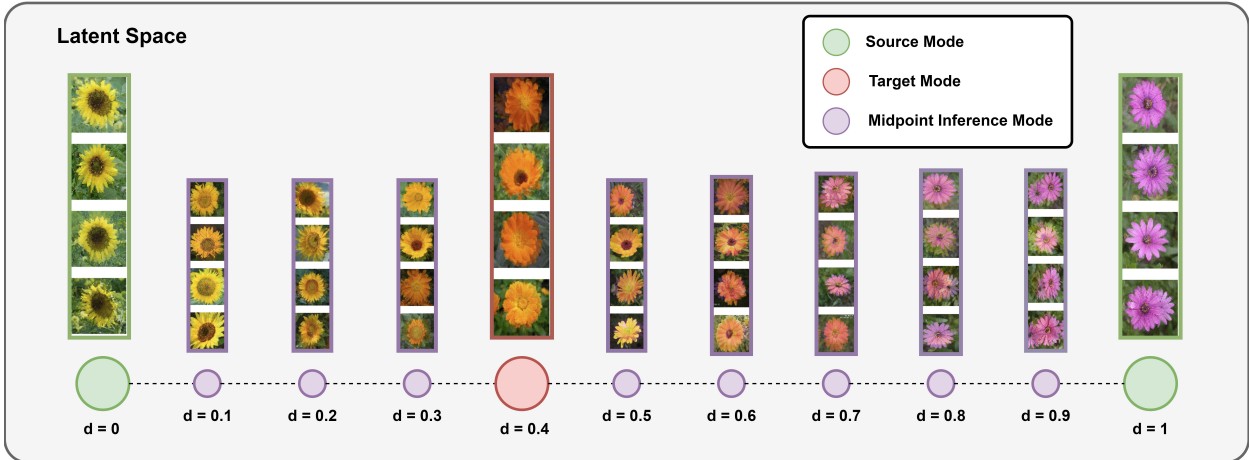

Figure 6: The generated image samples from the continual learning GAN model for tasks in the Oxford Flower dataset, with top-2 relevant source modes: (a) sunflower, located at $d = 0$, (b) osteospermum, located at $d = 1$, and the target mode: (c) orange dahlia, located at the coordinate $d = 0.4$. The midpoint inference samples are obtained by imputing the target and the source modes.

target task while preventing catastrophic forgetting. Specifically, we construct a target label embedding for each target based on the label embeddings of the top-2 closest learned modes and the computed affinity scores, as shown in Equation (3). Next, we fine-tune the GAN model with the newly labeled target samples, implementing generative replay to further avoid catastrophic forgetting of the existing modes. After incorporating the first target into the GAN model, we continue the continual learning process for the remaining targets. We compare our framework with sequential fine-tuning (Zhai et al., 2019), multi-task learning (Standley et al., 2020b), EWC-GAN (Seff et al., 2017), lifelong-GAN (Zhai et al., 2019), and CAM-GAN (Varshney et al., 2021) for the few-shot generative task with 100 target training samples. We report the FID scores for the average performance over all modes in Figure 4 (a) and the performance of the target mode in Figure 5 (a). By selectively choosing and utilizing the relevant knowledge from learned modes, our approach significantly outperforms the conventional training methods (i.e., sequential fine-tuning, and multi-task learning). The results also demonstrate that our proposed mode-aware continual learning approach significantly outperforms EWC-GAN (Seff et al., 2017) and achieves highly competitive performance in comparison to lifelong-GAN (Zhai et al., 2019) and CAM-GAN (Varshney et al., 2021). Although we observed a slight degradation in the performance of the top-2 closest modes due to the trade-off discussed in Theorem 1, our lifelong learning model demonstrates better overall performance when considering all the learned modes. In other words, the results demonstrate that our method improves overall performance as the model learns more tasks. Our method not only achieves gains in the target mode but also minimizes the loss in the learned modes, resulting in an improvement in average performance.

Next, we conducted similar experiments with the CIFAR-100 and other datasets. For CIFAR-100, we pretrained the model on the CIFAR-10 dataset. The model was then fine-tuned on six CIFAR-100 target tasks in a continual learning setting. Detailed information about the experiment setup is provided in Appendix B. The average performance and the target performance are presented in Figures 4 (b) and 5 (b), respectively. The results indicate a trend similar to the CIFAR-10 experiment, where our method consistently outperforms other methods throughout the learning process of the target tasks. To further evaluate the efficacy of our method on more challenging datasets, we constructed six additional target tasks from three distinct datasets: ImageNet, Flower, and CelebA. Specifically, Tasks 1 and 2 are from the ImageNet dataset (i.e., great white shark and German shepherd), Tasks 3 and 4 are from the Flower dataset (i.e., osteospermum and orange dahlia), and Tasks 5 and 6 are from the CelebA dataset (i.e., blonde and black hairs). In this experiment, our model was pretrained on the CIFAR-100 dataset. From the knowledge of CIFAR-100, the model then utilizes the fishes class to learn Tasks 1, the carnivores class for Task 2, the flowers class for Tasks 3 and 4, and the people class for Tasks 5 and 6. As shown in Figures 4 (c) and 5 (c), all methods faced challenges when

learning new tasks from different datasets. However, since our method efficiently leverages prior knowledge, we observed that after learning one task, our model performed significantly better when learning the next task from the same dataset. The target performance results indicate that our model adapts quickly to changes in datasets, helping to maintain the average performance over the continual learning process. While CAM-GAN achieved good performance, it did so at the cost of poorer performance on the target task.

Moreover, we evaluated the latent representation of the generative model by generating image samples from the midpoint inference modes. Specifically, we considered the orange dahlia class in the Oxford Flower dataset as the target. Using the mode affinity score, our framework identified the two closest modes to the target: the sunflower and osteospermum tasks from the previous experiment. Leveraging the knowledge from these related tasks, we formulated the target embedding label. The latent embedding space is illustrated in Figure 6, where the source tasks are positioned at $d = 0$ for the sunflower mode and $d = 1$ for the osteospermum mode. The target embedding is located at $d = 0.4$, indicating that the target is closer to the sunflower than to the osteospermum. The model was then fine-tuned with the target data, and the generated samples are shown at $d = 0.4$ in Figure 6. The results demonstrate that the model effectively leveraged the inherent similarity between the sunflower and osteospermum to enhance its ability to generate orange dahlia flowers. Additionally, we considered the midpoints between the source modes and the target, using the source and target modes to interpolate these midpoints. The generated samples indicate a smooth transition between the related modes in the model. It's worth noting that in some instances, the model may generate mode-collapsed samples. This occurrence can be attributed to the remarkably close resemblance between these two types of flowers. Overall, the results highlight the model's ability to utilize related modes efficiently, thereby improving its performance in generating target images by leveraging the similarities between the source and target tasks.

## 6 Ablation Studies

### 6.1 Mode-Aware Continual Learning

We apply the computed mode-affinity scores between generative tasks in the MNIST, CIFAR-10, and CIFAR-100 datasets to the mode-aware continual learning framework. In each dataset, we define two target tasks for continual learning scenarios and consider the remaining eight classes as source tasks. Particularly, (digit 0, digit 1), (truck, cat), and (lion, bus) are the targets for the MNIST, CIFAR-10, and CIFAR-100 experiments, respectively. The GAN model is trained to sequentially update these target tasks. Here, we select the top-2 closest modes to each target and leverage their knowledge for quick adaptation in learning the target task while preventing catastrophic forgetting. First, we construct a label embedding for the target data samples based on the label embeddings of the top-2 closest modes and the computed distances, as shown in Equation (3). Next, we fine-tune the source GAN model with the newly-labeled target samples, while also implementing memory replay to avoid catastrophic forgetting of the existing modes. After adding the first target to GAN, we continue the continual learning process for the second target in each experiment. We compare our framework with sequential fine-tuning (Zhai et al., 2019), multi-task learning (Standley et al., 2020b), LwF (Li & Hoiem, 2017), EWC-GAN (Seff et al., 2017), lifelong-GAN (Zhai et al., 2019), CAM-GAN (Varshney et al., 2021), and StyleCL (Kappiyath et al., 2025) for the few-shot generative task with 100 target data samples. We report the FID scores of the images from the target mode, top-2 closest modes, and the average of all modes in Table 2.

By selectively choosing and utilizing the relevant knowledge from learned modes, our approach significantly outperforms the conventional training methods (i.e., sequential fine-tuning, and multi-task learning) for both the first (i.e., digit 0, truck, lion) and the second generative tasks (i.e., digit 1, cat, and bus). The results further demonstrate that our proposed mode-aware continual learning approach significantly outperforms EWC-GAN (Seff et al., 2017) in the second target task in all datasets. Moreover, our model also achieves highly competitive results in comparison to lifelong-GAN (Zhai et al., 2019) and CAM-GAN (Varshney et al., 2021), showcasing its outstanding performance on the first and second target tasks. Although we observed a slight degradation in the performance of the top-2 closest modes due to the trade-off discussed in Theorem 1, our lifelong learning model demonstrates better overall performance when considering all the learned modes.

Table 2: Comparison of the continual learning frameworks for GAN against other baseline and state-of-the-art approaches for MNIST, CIFAR-10, and CIFAR-100, in terms of FID.

| | | MNIST | | |
|---|---|---|---|---|
| **Approach** | **Target** | $\mathcal{P}_{target}$ | $\mathcal{P}_{closest}$ | $\mathcal{P}_{average}$ |
| Sequential Fine-tuning (Zhai et al., 2019) | Digit 0 | 16.72 | 26.53 | 26.24 |
| Multi-task Learning (Standley et al., 2020b) | Digit 0 | 11.45 | **5.83** | 6.92 |
| LwF (Li & Hoiem, 2017) | Digit 0 | 9.23 | 7.85 | 8.32 |
| EWC-GAN (Seff et al., 2017) | Digit 0 | 8.96 | 7.51 | 7.88 |
| Lifelong-GAN (Zhai et al., 2019) | Digit 0 | 8.65 | 6.89 | 7.37 |
| CAM-GAN (Varshney et al., 2021) | Digit 0 | 7.02 | 6.43 | 6.41 |
| StyleCL (Kappiyath et al., 2025) | Digit 0 | 6.35 | **6.16** | 6.21 |
| **MA-Continual Learning-no replay (ours)** | Digit 0 | **6.18** | 6.73 | 6.67 |
| **MA-Continual Learning (ours)** | Digit 0 | 6.32 | 5.93 | **5.72** |
| EWC-GAN (Seff et al., 2017) | Digit 1 | 9.62 | 8.65 | 8.23 |
| Lifelong-GAN (Zhai et al., 2019) | Digit 1 | 8.74 | 7.31 | 7.29 |
| CAM-GAN (Varshney et al., 2021) | Digit 1 | 7.42 | 6.58 | 6.43 |
| **MA-Continual Learning (ours)** | Digit 1 | **6.45** | **6.14** | **5.92** |
| | | CIFAR-10 | | |
| **Approach** | **Target** | $\mathcal{P}_{target}$ | $\mathcal{P}_{closest}$ | $\mathcal{P}_{average}$ |
| Sequential Fine-tuning (Zhai et al., 2019) | Truck | 61.52 | 65.18 | 64.62 |
| Multi-task Learning (Standley et al., 2020b) | Truck | 55.32 | **33.65** | 35.52 |
| LwF (Li & Hoiem, 2017) | Truck | 45.95 | 38.35 | 37.48 |
| EWC-GAN (Seff et al., 2017) | Truck | 44.61 | 35.54 | 35.21 |
| Lifelong-GAN (Zhai et al., 2019) | Truck | 41.84 | 35.12 | 34.67 |
| CAM-GAN (Varshney et al., 2021) | Truck | 37.41 | 34.67 | 34.24 |
| StyleCL (Kappiyath et al., 2025) | Truck | 36.24 | 33.63 | 34.11 |
| **MA-Continual Learning-no replay (ours)** | Truck | **35.31** | 35.83 | 34.22 |
| **MA-Continual Learning (ours)** | Truck | 35.57 | 34.68 | **33.89** |
| EWC-GAN (Seff et al., 2017) | Cat | 45.17 | 36.53 | 35.62 |
| Lifelong-GAN (Zhai et al., 2019) | Cat | 42.58 | 35.76 | 34.89 |
| CAM-GAN (Varshney et al., 2021) | Cat | 37.29 | 35.28 | 34.62 |
| **MA-Continual Learning (ours)** | Cat | **35.29** | 34.76 | **34.01** |
| | | CIFAR-100 | | |
| **Approach** | **Target** | $\mathcal{P}_{target}$ | $\mathcal{P}_{closest}$ | $\mathcal{P}_{average}$ |
| Sequential Fine-tuning (Zhai et al., 2019) | Lion | 63.78 | 66.56 | 65.82 |
| Multi-task Learning (Standley et al., 2020b) | Lion | 56.32 | **36.38** | 37.47 |
| LwF (Li & Hoiem, 2017) | Lion | 47.41 | 39.26 | 38.13 |
| EWC-GAN (Seff et al., 2017) | Lion | 46.53 | 38.79 | 36.72 |
| Lifelong-GAN (Zhai et al., 2019) | Lion | 43.57 | 38.35 | 36.53 |
| CAM-GAN (Varshney et al., 2021) | Lion | 40.24 | 37.64 | 36.86 |
| StyleCL (Kappiyath et al., 2025) | Lion | 39.95 | 36.51 | 36.48 |
| **MA-Continual Learning-no replay (ours)** | Lion | **38.54** | 38.14 | 36.72 |
| **MA-Continual Learning (ours)** | Lion | 38.73 | 36.53 | **35.88** |
| EWC-GAN (Seff et al., 2017) | Bus | 49.86 | 39.84 | 37.91 |
| Lifelong-GAN (Zhai et al., 2019) | Bus | 43.73 | 39.75 | 37.66 |
| CAM-GAN (Varshney et al., 2021) | Bus | 42.81 | 38.82 | 37.21 |
| **MA-Continual Learning (ours)** | Bus | **41.68** | 38.63 | **36.87** |

Table 3: Comparison of the mode-aware continual learning performance between different choices of the number of closest modes

| Approach | Dataset | Target | Performance |
|---|---|---|---|
| MA-Continual Learning with top-2 closest modes | MNIST | Digit 0 | 6.32 |
| **MA-Continual Learning with top-3 closest modes** | **MNIST** | **Digit 0** | **6.11** |
| MA-Continual Learning with top-4 closest modes | MNIST | Digit 0 | 6.78 |
| MA-Continual Learning with all modes | MNIST | Digit 0 | 8.36 |
| MA-Continual Learning with top-2 closest modes | CIFAR-10 | Truck | 35.57 |
| **MA-Continual Learning with top-3 closest modes** | **CIFAR-10** | **Truck** | **35.52** |
| MA-Continual Learning with top-4 closest modes | CIFAR-10 | Truck | 36.31 |
| MA-Continual Learning with all modes | CIFAR-10 | Truck | 45.92 |
| MA-Continual Learning with top-2 closest modes | CIFAR-100 | Lion | 38.73 |
| MA-Continual Learning with top-3 closest modes | CIFAR-100 | Lion | 38.54 |
| **MA-Continual Learning with top-4 closest modes** | **CIFAR-100** | **Lion** | **38.31** |
| MA-Continual Learning with all modes | CIFAR-100 | Lion | 48.55 |

## 6.2 Choice of closest modes

In this experiment, we evaluate the effectiveness of our proposed continual learning framework by varying the number of closest existing modes used for fine-tuning the target mode. Throughout this paper, we opt to utilize the top-2 closest modes, a choice driven by its minimal computational requirements. Opting for a single closest mode (i.e., transfer learning scenarios) would essentially replace that mode with the target mode, negating the concept of continual learning. Here, we explore different scenarios across the MNIST, CIFAR-10, and CIFAR-100 datasets, where we investigate the top-2, top-3, and top-4 closest modes for continual learning. As detailed in Table 3, selecting the three closest modes yields the most favorable target generation performance in the MNIST and CIFAR-10 experiments. Notably, knowledge transfer from the four closest modes results in the weakest performance. This discrepancy can be attributed to the simplicity of these datasets and their highly distinguishable data classes. In such cases, employing more tasks resembles working with dissimilar tasks, leading to negative transfer during target mode training. Conversely, in the CIFAR-100 experiment, opting for the top-4 modes yields the best performance. This outcome stems from the dataset's complexity, where utilizing a larger set of relevant modes confers an advantage during the fine-tuning process. In summary, the choice of the top-N closest modes is highly dependent on the dataset and available computational resources. Employing more modes necessitates significantly more computational resources and training time for memory replay of existing tasks. It's crucial to note that with an increased number of related modes, the model requires more time and data to converge effectively.

## 7 Conclusion and Limitations

We present a new measure of similarity between generative tasks for GANs. This measure provides insight into the difficulty of extracting valuable knowledge from existing modes to learn new tasks. We apply this metric within the continual learning framework, capitalizing on the knowledge acquired from relevant learned modes to expedite adaptation to new target modes. Through various experiments, we empirically validate the efficacy of our approach, highlighting its advantages over traditional fine-tuning methods and other state-of-the-art continual learning techniques.

One notable constraint within our continual learning framework lies in its dependence on pretraining the model with a relevant database. This database, which must contain knowledge related to the target, is essential for effectively guiding the learning process. In practical scenarios, we often encounter pretrained models that align closely with the target data or belong to similar categories. Leveraging these pretrained models directly within our framework greatly enhances their versatility and operational efficiency.

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

## A    Theoretical Analysis

We first recall the definition of the GAN's discriminator loss as follows:

**Definition 2** (Discriminator Loss). *Let $x = \{x_1, \ldots, x_m\}$ be the real data samples, $z$ denote the random vector, and $\theta_{\mathcal{D}}$ be the discriminator's parameters. $\mathcal{D}$ is trained to maximize the probability of assigning the correct label to both training real samples and generated samples $\mathcal{G}(z)$ from the generator $\mathcal{G}$. The objective of the discriminator is to maximize the following function:*

$$\nabla_{\theta_{\mathcal{D}}} \sum_{i=1}^{m} \left[ \log \mathcal{D}\left(x^{(i)}\right) + \log\left(1 - \mathcal{D}\left(\mathcal{G}\left(z^{(i)}\right)\right)\right) \right] \tag{5}$$

We recall the definition of Fisher Information matrix (FIM) (Le et al., 2022a) as follows:

**Definition 3** (Fisher Information). *Given dataset $X$, let $N$ denote a neural network with weights $\theta$, and the negative log-likelihood loss function $L(\theta) := L(\theta, X)$. FIM is described as follows:*

$$F(\theta) = \mathbb{E}\left[\nabla_\theta L(\theta) \nabla_\theta L(\theta)^T\right] = -\mathbb{E}\left[H\left(L(\theta)\right)\right] \tag{6}$$

Next, we present the proof of Theorem 1.

**Theorem 1.** *Let $X_a$ be the source data, characterized by the density function $p_a$. Let $X_b$ be the data for the target mode with data density function $p_b$, $p_b \neq p_a$. Let $\theta$ denote the model's parameters. Consider the loss functions $L_a(\theta) = \mathbb{E}[l(X_a; \theta)]$ and $L_b(\theta) = \mathbb{E}[l(X_b; \theta)]$. Assume that both $L_a(\theta)$ and $L_b(\theta)$ are strictly convex and possess distinct global minima. Let $X_n$ denote the mixture data of $X_a$ and $X_b$ described by $p_n = \alpha p_a + (1 - \alpha)p_b$, where $\alpha \in (0, 1)$. The corresponding loss function is given by $L_n(\theta) = \mathbb{E}[l(X_n; \theta)]$. Under these assumptions, it follows that $\theta^* = \arg\min_\theta L_n(\theta)$ satisfies:*

$$L_a(\theta^*) > \min_\theta L_a(\theta) \tag{7}$$

**Proof of Theorem 1**. Assume toward contradiction that $L_a(\theta^*) > \min_\theta L_a(\theta)$ does not hold. Because $L_a(\theta^*) \geq \min_\theta L_a(\theta)$ always holds, we must have that:

$$L_a(\theta^*) = \min_\theta L_a(\theta). \tag{8}$$

By the linearity of expectation, we have that:

$$L_n(\theta) = \alpha L_a(\theta) + (1-\alpha)L_b(\theta)$$

Hence, we have

$$
\begin{aligned}
\min_\theta L_n(\theta) = L_n(\theta^*) \\
&= \alpha L_a(\theta^*) + (1-\alpha)L_b(\theta^*) \\
&= \alpha \min_\theta L_a(\theta) + (1-\alpha)L_b(\theta^*) \\
&> \alpha \min_\theta L_a(\theta) + (1-\alpha)L_a(\theta^*) \\
&= \alpha \min_\theta L_a(\theta) + (1-\alpha)\min_\theta L_a(\theta) \\
&= \min_\theta L_a(\theta)
\end{aligned}
$$

where in the third equality we use the facts that both $L_a$ and $L_b$ are strongly convex and have different global minimums. Because $L_a$ and $L_b$ have the same optimal value (assumed to be 0) and that $\theta^*$ is not the optimal point for $L_b$, we must have $L_b(\theta^*) > L_b(\theta_b^*) = L_a(\theta^*)$ where $\theta_b^* = \arg\min_\theta L_b(\theta)$.

Therefore, we have proved that $\min_\theta L_n(\theta) = L_n(\theta^*) > \min_\theta L_a(\theta)$, contradicting to Eq. equation 8. $\qquad\square$

## B  Experimental Setup

In this work, we construct 40 generative tasks based on popular datasets such as MNIST, CIFAR-10, CIFAR-100, ImageNet, Oxford Flower, and CelebA. For MNIST, we define 10 distinct generative tasks, each focused on generating a specific digit (i.e., $0, 1, \ldots, 9$). Task 0, for example, is designed to generate the digit 0, while task 1 generates the digit 1, and so on. For the CIFAR-10 dataset, we also construct 10 generative tasks, with each task aimed at generating a specific object category such as airplane, automobile, bird, cat, deer, dog, frog, horse, ship, and truck. Similarly, for the CIFAR-100 dataset, we create 10 target tasks, each corresponding to a specific image class, including bear, leopard, lion, tiger, wolf, bus, pickup truck, train, streetcar, and tractor. For ImageNet, we create 2 target tasks, each corresponding to a specific image class, including the great white shark and the German shepherd. In Oxford Flower dataset, we also pick 2 target tasks, which correspond to osteospermum and orange dahlia. Each flower category consists of 80 image samples. The sample was originally $128 \times 128$, but resized to $32 \times 32$ to reduce the computational complexity. Lastly, we choose 2 target tasks in the CelebA dataset, which correspond to the image class blond hair and black hair.

To represent the generative tasks, we utilize the conditional Wasserstein GAN with Gradient Penalty (WGAN-GP) model (Gulrajani et al., 2017; Yonekura et al., 2021). In each experiment, we select a specific task as the target task, while considering the other tasks as source tasks. To represent these source tasks, we train the WGAN-GP model on their respective datasets. This enables us to generate high-quality samples that are representative of the source tasks. Once trained, we can use the WGAN-GP model as the representation network for the generative tasks. This model is then applied to our proposed mode-aware continual learning framework. We compare our method against several approaches, including individual learning (Mirza & Osindero, 2014), sequential fine-tuning (Wang et al., 2018), multi-task learning (Standley et al., 2020b), EWC-GAN (Seff et al., 2017), Lifelong-GAN (Zhai et al., 2019), and CAM-GAN (Varshney et al., 2021). Individual learning (Mirza & Osindero, 2014) involves training the GAN model on a specific task in isolation. In sequential fine-tuning (Wang et al., 2018), the GAN model is trained sequentially on source and target tasks. Multi-task learning (Standley et al., 2020b), on the other hand, involves training

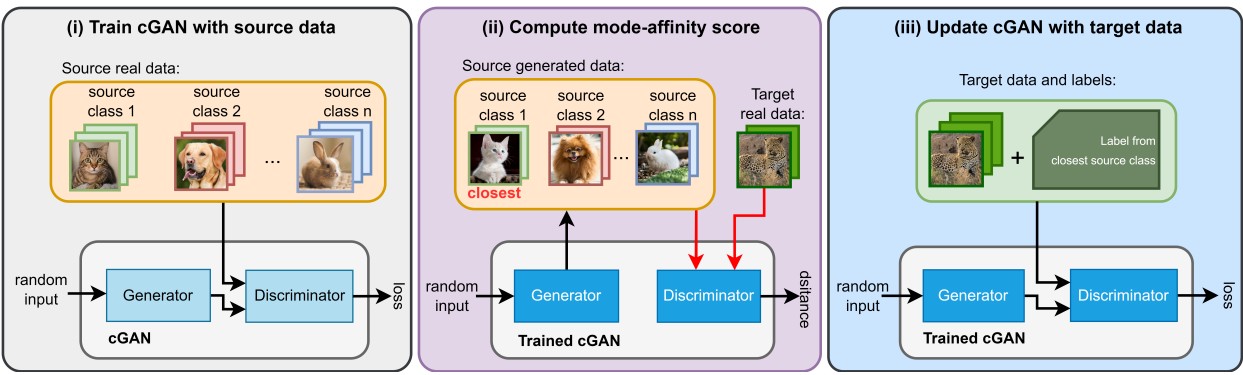

Figure 7: The overview of mode-aware transfer learning framework for the conditional Generative Adversarial Network: (i) Representing source data classes using GAN, (ii) Computing the mode-affinity from each source mode to the target, (iii) Fine-tuning the generative model using the target data and the label of the closest mode for transfer learning.

---

**Algorithm 2:** Mode-Aware Transfer Learning for Conditional Generative Adversarial Networks

**Data:** Source data: $(X_{train}, y_{train})$, Target data: $X_{target}$
**Input:** The generator $\mathcal{G}$ and discriminator $\mathcal{D}$ of GAN
**Output:** Target generator $\mathcal{G}_{\bar{\theta}}$
**Function** dMAS($X_a, y_a, X_b, \mathcal{G}, \mathcal{D}$):

  Generate data $\tilde{X}_a$ of class label $y_a$ using the generator $\mathcal{G}$

  Compute $H_a$ from the loss of discriminator $\mathcal{D}$ using $\{X_a, \tilde{X}_a\}$

  Compute $H_b$ from the loss of discriminator $\mathcal{D}$ using $\{X_b, \tilde{X}_a\}$

  **return** $s[a,b] = \dfrac{1}{\sqrt{2}} \left\| H_a^{1/2} - H_b^{1/2} \right\|_F$

**Function** Main:

  Train $(\mathcal{G}_\theta, \mathcal{D}_\theta)$ with $X_{train}, y_{train}$                          ▷ Pre-train GAN model
  Construct S source modes, each from a data class in $y_{train}$
  **for** $i = 1, 2, \ldots, S$ **do**
    $s_i = $ dMAS($X_{train_i}, y_{train_i}, X_{target}, \mathcal{G}_\theta, \mathcal{D}_\theta$)                ▷ Find the closest modes
  **return** closest mode(s): $i^* = \underset{i}{\operatorname{argmin}} \ s_i$

                                                                                        ▷ Fine-tune with the target task

  **while** $\theta$ *not converged* **do**
    Update $\mathcal{G}_\theta, \mathcal{D}_\theta$ using real data $X_{target}$ and closest source label $y_{train_{i^*}}$
  **return** $\mathcal{G}_{\bar{\theta}}$

---

a GAN model on a joint dataset created from both the source and target tasks. Our method is designed to improve on these approaches by enabling the continual learning of generative tasks while mitigating catastrophic forgetting.

## C   Mode-Aware Transfer Learning

We apply the proposed mode-affinity score to transfer learning in an image generation scenario. The proposed similarity measure enables the identification of the closest modes or data classes to support the learning of the target mode. Here, we introduce a *mode-aware transfer learning* framework that quickly adapts a pretrained GAN model to learn the target mode. The overview of the transfer learning framework is illustrated in Figure 7. Particularly, we select the closest source mode from the pool of multiple learned modes based on the computed dMAS.

Table 4: Comparison of the mode-aware transfer learning framework for GAN against other baselines and FID-transfer learning approach in terms of FID.

| Approach | Target | MNIST 10-shot | 20-shot | 100-shot |
|---|---|---|---|---|
| Individual Learning (Mirza & Osindero, 2014) | Digit 0 | 34.25 | 27.17 | 19.62 |
| Sequential Fine-tuning (Zhai et al., 2019) | Digit 0 | 29.68 | 24.22 | 16.14 |
| Multi-task Learning (Standley et al., 2020b) | Digit 0 | 26.51 | 20.74 | 10.95 |
| FID-Transfer Learning (Wang et al., 2018) | Digit 0 | **12.64** | **7.51** | **5.53** |
| **MA-Transfer Learning (ours)** | **Digit 0** | **12.64** | **7.51** | **5.53** |
| Individual Learning (Mirza & Osindero, 2014) | Digit 1 | 35.07 | 29.62 | 20.83 |
| Sequential Fine-tuning (Zhai et al., 2019) | Digit 1 | 28.35 | 24.79 | 15.85 |
| Multi-task Learning (Standley et al., 2020b) | Digit 1 | 26.98 | 21.56 | 10.68 |
| FID-Transfer Learning (Wang et al., 2018) | Digit 1 | **11.35** | **7.12** | **5.28** |
| **MA-Transfer Learning (ours)** | **Digit 1** | **11.35** | **7.12** | **5.28** |
| Approach | Target | CIFAR-10 10-shot | 20-shot | 100-shot |
| Individual Learning (Mirza & Osindero, 2014) | Truck | 89.35 | 81.74 | 72.18 |
| Sequential Fine-tuning (Zhai et al., 2019) | Truck | 76.93 | 70.39 | 61.41 |
| Multi-task Learning (Standley et al., 2020b) | Truck | 72.06 | 65.38 | 55.29 |
| FID-Transfer Learning (Wang et al., 2018) | Truck | **51.05** | **44.93** | **36.74** |
| **MA-Transfer Learning (ours)** | **Truck** | **51.05** | **44.93** | **36.74** |
| Individual Learning (Mirza & Osindero, 2014) | Cat | 80.25 | 74.46 | 65.18 |
| Sequential Fine-tuning (Zhai et al., 2019) | Cat | 73.51 | 68.23 | 59.08 |
| Multi-task Learning (Standley et al., 2020b) | Cat | 68.73 | 61.32 | 50.65 |
| FID-Transfer Learning (Wang et al., 2018) | Cat | **47.39** | **40.75** | **32.46** |
| **MA-Transfer Learning (ours)** | **Cat** | **47.39** | **40.75** | **32.46** |
| Approach | Target | CIFAR-100 10-shot | 20-shot | 100-shot |
| Individual Learning (Mirza & Osindero, 2014) | Lion | 87.91 | 80.21 | 72.58 |
| Sequential Fine-tuning (Zhai et al., 2019) | Lion | 77.56 | 70.76 | 61.33 |
| Multi-task Learning (Standley et al., 2020b) | Lion | 71.25 | 67.84 | 56.12 |
| FID-Transfer Learning (Wang et al., 2018) | Lion | **51.08** | **46.97** | **37.51** |
| **MA-Transfer Learning (ours)** | **Lion** | **51.08** | **46.97** | **37.51** |
| Individual Learning (Mirza & Osindero, 2014) | Bus | 94.82 | 89.01 | 78.47 |
| Sequential Fine-tuning (Zhai et al., 2019) | Bus | 88.03 | 79.51 | 67.33 |
| Multi-task Learning (Standley et al., 2020b) | Bus | 80.06 | 76.33 | 61.59 |
| FID-Transfer Learning (Wang et al., 2018) | Bus | 61.34 | 54.18 | 46.37 |
| **MA-Transfer Learning (ours)** | **Bus** | **57.16** | **50.06** | **41.81** |

To leverage the knowledge of the closest mode for training the target mode, we assign the target data samples with labels of the closest mode. Subsequently, we use these modified target data samples to fine-tune the generator and discriminator of the pre-trained GAN model. Figure 7(3) illustrates the transfer learning method, where the data class 1 (i.e., cat images) is the most similar to the target data (i.e., leopard image) based on the computed dMAS. Hence, we assign the label of class 1 to the leopard images. The pre-trained GAN model uses this modified target data to quickly adapt the cat image generation to the leopard image generation. The mode-aware algorithm for transfer learning in GAN is described in Algorithm 2. By assigning the closest mode's label to the target data samples, our method can effectively fine-tune the relevant parts of GAN for learning the target mode. This approach helps improve the training process and reduces the number of required training data.

Next, we conduct experiments employing mode affinity scores within the context of transfer learning scenarios. These experiments were designed to assess the effectiveness of our proposed mode-affinity measure

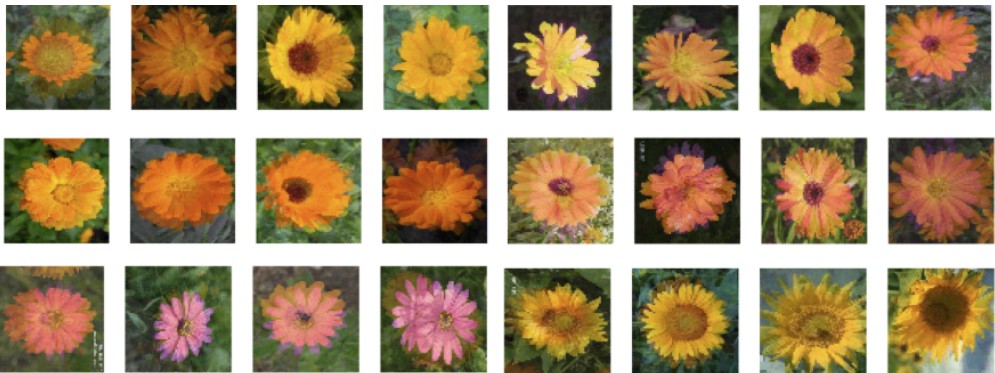

Figure 8: The generated image samples from the continual learning GAN model for tasks in the Oxford Flower dataset.

in the transfer learning framework. In this scenario, each generative task corresponds to a single data class within the MNIST (LeCun et al., 2010), CIFAR-10 (Krizhevsky et al., 2009), and CIFAR-100 (Krizhevsky et al., 2009) datasets. Here, in our transfer learning framework, we leverage the computed mode-affinity scores between generative tasks. Specifically, we utilize this distance metric to identify the mode closest to the target mode and then fine-tune the conditional Generative Adversarial Network (GAN) accordingly. To achieve this, we assign the target data samples with the labels of the closest mode and use these newly-labeled samples to train the GAN model. By doing so, the generative model can benefit from the knowledge acquired from the closest mode, enabling quick adaptation in learning the target mode. In this study, we compare our proposed transfer learning framework with several baselines and state-of-the-art approaches, including individual learning (Mirza & Osindero, 2014), sequential fine-tuning (Wang et al., 2018), multi-task learning (Standley et al., 2020b), and FID-transfer learning (Wang et al., 2018). Additionally, we present a performance comparison of our mode-aware transfer learning approach with these methods for 10-shot, 20-shot, and 100-shot scenarios in the MNIST, CIFAR-10, and CIFAR-100 datasets (i.e., the target dataset contains only 10, 20, or 100 data samples).

Across all three datasets, our results demonstrate the effectiveness of our approach in terms of generative performance and its ability to efficiently learn new tasks. Our proposed framework significantly outperforms individual learning and sequential fine-tuning while demonstrating strong performance even with fewer samples compared to multi-task learning. Moreover, our approach is competitive with FID transfer learning, where the similarity measure between generative tasks is based on FID scores. Notably, our experiments with the CIFAR-100 dataset reveal that FID scores may not align with intuition and often result in poor performance. Notably, for the MNIST dataset, we consider generating digits 0 and 1 as the target modes. As shown in Table 4, our method outperforms individual learning, sequential fine-tuning, and multi-task learning approaches significantly, while achieving similar results compared with the FID transfer learning method. Since the individual learning model lacks training data, it can only produce low-quality samples. On the other hand, the sequential fine-tuning and multi-task learning models use the entire source dataset while training the target mode, which results in better performance than the individual learning method. However, they cannot identify the most relevant source mode and data, thus, making them inefficient compared with our proposed mode-aware transfer learning approach. In other words, the proposed approach can generate high-quality images with fewer target training samples. Notably, the proposed approach can achieve better results using only 20% of data samples. For more complex tasks, such as generating cat and truck images in CIFAR-10 and lion and bus images in CIFAR-100, our approach achieves competitive results to other methods while requiring only 10% training samples. The mode-aware transfer learning framework using the Discriminator-based Mode Affinity Score can effectively identify relevant source modes and utilize their knowledge for learning the target mode.

## D    Computational Cost

As the model learns new tasks and the number of learned tasks increases, the computational cost of measuring the distance between incoming tasks and learned tasks will also grow. To mitigate this while preserving the accuracy of task distance, we propose maintaining a reference list of learned tasks that are the most unique. In other words, instead of tracking every learned task, we will only retain those that are distinct. When a new task arrives, we will only measure the distance from this new task to the reference tasks to decide the weighted label embedding for the new task.

A new task will be added to this reference list only if it has been learned correctly and its distance from existing reference tasks exceeds a fixed threshold. Conversely, if a task is too similar to an existing reference task, it will not be considered as a new reference task. Additionally, tasks will be removed from the reference list if they do not rank among the top five most relevant tasks for the next ten incoming tasks.

By keeping the reference list concise and unique, we can significantly reduce the computational cost of measuring task distances while ensuring the continued effectiveness of our approach.

