# OpenReview forum: "Mode-Aware GAN: Continual Adaptation for Conditional Image Generation"
_TMLR — Rejected by TMLR_

### Review · Reviewer_pC4J · 2025-03-18

**Summary Of Contributions:**

This paper proposes a novel mode-aware continual learning framework, introducing a mode-affinity score (dMAS) to help Generative Adversarial Networks (GANs) effectively avoid catastrophic forgetting in continual learning and quickly adapt to new tasks. Experimental validation on multiple datasets demonstrates that the method outperforms traditional approaches in reducing training sample requirements and improving generative performance.

**Audience:**

Yes

**Broader Impact Concerns:**

The paper does not have any apparent ethical issues.

**Claims And Evidence:**

Yes

**Requested Changes:**

1. Review some of the latest state-of-the-art methods and make a comprehensive comparison with them.
2. Provide more qualitative comparison results based on different datasets.
3. Please address the issues raised in **Weaknesses**.

**Strengths And Weaknesses:**

**Strengths:**
1. This paper aims to generate target images with a few training samples while retaining previously acquired knowledge from earlier tasks, which is an interesting and challenging problem.
2. The proposed method is inspiring and shows competitive performance in comparison with previous approaches.
3. Extensive experiments have been conducted to support their claims.

**Weaknesses:**
1. The related works and baselines presented in this paper are all from before 2022. Has there been no further research on this topic in recent years?
2. Why not present some qualitative results (generated images) to provide a more intuitive comparison of the performance of this method and previous methods in continual learning?
3. It seems that this method is only suitable for situations where the new task is similar to the previously learned tasks. Would it still perform well if there is a significant difference between the new task and the learned tasks? For example, a pretrained GAN can only generate animals, while the new task requires generating rockets.
4. The calculation of dMAS may lead to high computational cost, especially when the number of modes increases.

---

> ### Author Response · Authors · 2025-03-31
> **Response to Reviewer pC4J**
>
> Thank you very much for your detailed review. Your suggestions and concerns have been invaluable in improving our paper. We have incorporated your feedback in the latest revision (uploaded above) and would like to address your remaining concerns below:
>
> * Our continuous learning approach in GAN, which incorporates embedding labels for new tasks, is unique, and only a few papers have addressed this continual learning issue. We have compared our method with the most relevant approaches, including CAM-GAN and Lifelong-GAN. However, we are open to including additional comparable methods, and we would greatly appreciate any suggestions you may have for relevant papers.
>
> * We have included additional generated figures in the revised version. Initially, we limited the number of figures due to page constraints.
>
> * You are correct that our method relies heavily on the similarity between learned tasks and new tasks. This mirrors how humans acquire knowledge—progressing step by step. For example, a student must learn Calculus 1 before taking Calculus 2 and cannot enroll in an advanced medical course without first studying foundational Chemistry and Biology. Similarly, our model requires an initial database to establish foundational knowledge, enabling it to efficiently learn new, relevant tasks.
>
> * As you mentioned, the computational cost of task distance measurement (dMAS) increases as more tasks are learned. To address this, we have added a discussion in the Appendix of our revised paper, outlining a strategy to mitigate this issue while preserving task distance accuracy. Specifically, we propose maintaining a reference list of the most unique learned tasks rather than storing all learned tasks. A new task is added to this reference list only if it has been learned perfectly and its distance from existing reference tasks exceeds a fixed threshold. If a task is too similar to an existing reference task, it is not added. Additionally, tasks are removed from the reference list if they do not rank among the top five most relevant tasks for the next ten incoming tasks. By keeping the reference list concise and unique, we significantly reduce computational costs while maintaining the effectiveness of our approach.
>
> We truly appreciate your insightful feedback and look forward to any additional suggestions you may have.

---

> > ### Comment · Reviewer_pC4J · 2025-04-04
> > **Response**
> >
> > I appreciate the authors' rebuttals, which have addressed my concerns.

---

> > > ### Author Response · Authors · 2025-05-12
> > > **Official Comment by Authors**
> > >
> > > Thank you for your thoughtful consideration. If you have any other concerns please let us know.

---

### Review · Reviewer_kGiA · 2025-04-02

**Summary Of Contributions:**

The paper focus on continually learning new models in generative models over time while retaining the knowledge from previous tasks. The authors propose a discriminator-based mode affinity score to measure the similarity between different generative tasks. Based on the affinity scores, the modes from previously learned tasks which are closest to the target task are identified and integrated into the new mode for quick adaptation of new target data. In addition to this, the authors use generative replay to prevent forgetting and achieves competitive results on several datasets.

**Audience:**

Yes

**Claims And Evidence:**

No

**Requested Changes:**

I would request the authors to address the four points in weaknesses mentioned above.

**Strengths And Weaknesses:**

Strengths -
1. The paper is well written with a good motivation, illustrations, pseudo-code, and extensive experiments and ablations.
2. The author propose a simple metric of frechet distance hessian matrices as the mode affinity score.
3. The method uses a combination of the embeddings of closest models to approximate an embedding for the target data.
4. The experiments on consistency of mode affinity score is appreciated.

Weaknesses-
1. It is difficult to read figure 4 and 5 without any y-axis label. It looks like this is not the average performance. Poorly written captions could be improved.
2. How does the standard CL method LwF perform here. This could be a relevant and strong baseline similar to EWC-GAN which is used here. The compared methods are not very recent works. More recent works could be included.
3. Some important ablations of the proposed method are missing: what is the contribution of using generative replay? How dependent is the mode aware GAN on the generative replay method? Some more details of the generative replay method could be added.
4. Instead of taking the combination of closest modes, how about combining all previous modes with weightage depending on affinity scores? This could be interesting to see.

---

> ### Author Response · Authors · 2025-04-10
> **Response to Reviewer kGiA**
>
> Thank you very much for your detailed feedback. We have revised the manuscript accordingly and would like to address your remaining concerns in detail below:
>
> * We have added the y-axis label (i.e., Performance (%)) in the revised manuscript. Additionally, we updated the caption of Figure 4 to clarify that it presents the performance across all learned tasks.
>
> * We have added the standard LwF baseline in Table 2. Our method is now compared with the most relevant approaches, including CAM-GAN and Lifelong-GAN. We are open to incorporating additional comparable methods and would greatly appreciate any further suggestions you may have for relevant papers.
>
> * We have included the model’s performance without replay in Table 2. As expected, without replay, the model exhibits noticeable forgetting, particularly in tasks most similar to the target task. This is reasonable, as those tasks share more parameters with the target and are more likely to be affected during fine-tuning. Our simple replay strategy—generating images from the model itself—helps retain knowledge across all previously learned tasks.
>
> * In Table 3, we compare the model’s performance when using all learned tasks versus only the top-N most relevant tasks. The results show that performance gains diminish as non-relevant tasks are added. This highlights the importance of selecting only relevant tasks to facilitate positive knowledge transfer.
>
> We truly appreciate your insightful feedback and look forward to any additional suggestions you may have.

---

> > ### Comment · Reviewer_kGiA · 2025-05-11
> >
> > I would like to thank the authors for their response and my concerns are addressed in the current version of the paper.

---

> > > ### Author Response · Authors · 2025-05-12
> > > **Official Comment by Authors**
> > >
> > > Thank you for your thoughtful consideration. Should you have any further questions or suggestions, we would be glad to address them.

---

### Review · Reviewer_GtFb · 2025-04-21

**Summary Of Contributions:**

The paper proposes a continual learning framework for generative adversarial networks. It computes a mode affinity score between the target and the closest source modes. The most relevant mode information is used for knowledge transfer to enable quick adaptation to the target data. The framework adds a new mode to the generative model to represent the target task. This new mode is assigned an embedding label derived from the embeddings of the closest modes and the computed dMAS values between the closest modes and the target. The closest affinity scores are used to enhance knowledge transfer, while forgetting is mitigated using a generative replay approach. The results are demonstrated on various standard datasets.

**Audience:**

Yes

**Broader Impact Concerns:**

Broader Impact is not discussed, I request the author please add the same.

**Claims And Evidence:**

No

**Requested Changes:**

Please refer to the **Strengths And Weaknesses** section.

**Strengths And Weaknesses:**

**Strengths:**

1: The paper addresses the challenging problem of continual learning in the generative domain and proposes a simple yet effective solution.

2:  Experimental results are reported on standard datasets such as CIFAR-10, CIFAR-100, Flowers, ImageNet, and CelebA, demonstrating improvements over baseline models.

3: The idea of computing a Mode Affinity Score for knowledge transfer is potentially interesting and novel.

 4: The consistent performance improvements across various datasets and evaluation metrics highlight the effectiveness of the proposed approach.


**Weaknesses:**

1: The Mode Affinity Score leverages Fisher information from the discriminator's loss with respect to input samples. A similar idea has already been explored in CAM-GAN. The paper does not clearly explain how its approach differs from CAM-GAN, which raises concerns about novelty.

2: In the continual learning (CL) setting, access to source data during adaptation is typically restricted. However, in Algorithm 1, the presence of `X_source` and `Y_source` for adapting to `X_target` appears to violate the CL setting. This contradiction needs clarification.

3: The motivation behind the construction of `emb(Y_target)` is unclear. It seems that the embedding of the closest class is used as an initialization during the target model adaptation, which helps due to its proximity to the target mode. However, the approach to extending this to multiple classes is not explained. For example, how would the method handle a scenario where the target consists of 10 new classes? The paper needs to clarify how `emb(Y_target)` is generated for such multi-class scenarios.

4: Algorithm 1 lacks clarity. Both the generator and discriminator are denoted by the same parameter symbol, $\theta$, which is confusing. It should be explicitly stated which parameters are being optimized and how. Moreover, it is not clear how `emb(Y_target)` is updated, since it is not part of the model parameters. The algorithm needs to be rewritten with clear and well-defined steps.

5: The paper does not include any recent baselines. The latest baseline used is CAM-GAN (2021), which is 3–4 years old. It is strongly recommended that the authors include and compare against more recent works such as:

   - [a] LFS-GAN: Lifelong Few-Shot Image Generation, ICCV 2023
   - [b] Task-Free Continual Generation and Representation Learning via Dynamically Expandable Memory Cluster, AAAI 2024
   - [c] Lifelong Learning in StyleGAN through Latent Subspaces, TMLR 2025

6: The paper lacks a strong motivation for the proposed method. It mainly focuses on solving the problem without providing sufficient reasoning or justification behind key design choices. Furthermore, the writing quality needs significant improvement—the paper lacks a logical flow, making it difficult to understand.

7: The paper often considers each class as a separate task, which is impractical in real-world scenarios. Treating each class as an individual task can lead to thousands of tasks, which is not scalable or manageable within a generative replay framework.

8: The affinity score is essentially used to find the closest classes. This raises the question: why not use a pre-trained image classification model to identify the closest classes and extract their embeddings directly? This could be a simpler and more effective alternative.

---

> ### Author Response · Authors · 2025-04-28
> **Response to Reviewer GtFb**
>
> Thank you for your thorough feedback. We have updated the manuscript accordingly and provide detailed responses to your concerns below.
>
> 1. > "The Mode Affinity Score leverages Fisher information from the discriminator's loss with respect to input samples. A similar idea has already been explored in CAM-GAN...
>
> In CAM-GAN, the task similarity measure is based on the generator loss, which is expensive to compute directly and thus must be approximated using the convolutional layers in the adapter module. In contrast, our proposed method measures task similarity using the discriminator loss, which is significantly simpler to compute and has been shown to be reliable, as demonstrated in Figure 2.
>
> 2. > "In the continual learning (CL) setting, access to source data during adaptation is typically restricted. However, in Algorithm 1, the presence of X_source and Y_source for adapting to X_target appears to violate the CL setting..."
>
> Thank you for pointing out the typo. We have corrected the algorithm accordingly. In our method, we measure the distance from the source task to the target task using data generated by the source generator G and real data from the target task.
>
> 3. >  "The motivation behind the construction of emb(Y_target) is unclear..."
>
> When a task contains multi-class data, the model computes the distance between each class of the new data and the previously learned source classes. Based on these distances, it assigns weighted labels to the new data and updates the model to learn the new classes while maintaining performance on the existing ones.
>
> 4. > "Algorithm 1 lacks clarity. Both the generator and discriminator are denoted by the same parameter symbol..."
>
> Thank you for highlighting these issues. We have revised the algorithm according to your suggestions.
>
> 5. > "The paper does not include any recent baselines..."
>
> Thank you for introducing us to these recent breakthroughs. We have added a comparison with [c] "Lifelong Learning in StyleGAN through Latent Subspaces," TMLR 2025, in the revised manuscript. [a] is a few-shot learning method, and [b] focuses on continual learning for VAE models. While both [a] and [b] are excellent contributions, they are not directly comparable to our work, which addresses continual learning specifically for GAN models.
>
> 6. > "The paper lacks a strong motivation for the proposed method..."
>
> In existing continual learning methods for GANs, such as CAM-GAN, Lifelong-GAN, and StyleCL, the goal is to learn from a continuously arriving data stream, with the number of data classes known in advance. In other words, if new data arrives after the initial stream, these models cannot handle it because the corresponding labels were not predefined.
>
> In our approach, we first construct a large, well-trained GAN model that has already learned a number of data classes. When new data arrives, our model dynamically assigns new labels based on the existing label structure and trains accordingly while preserving knowledge of the previously learned classes. The model continuously learns new classes one by one, similar to the setup in other continual learning frameworks.
>
> 7. > "The paper often considers each class as a separate task, which is impractical in real-world scenarios..."
>
> In this paper, we address the problem of continually learning to generate new data classes using GANs. Inspired by how humans learn, we initialize the model with a group of source tasks, treating them as base knowledge. When a new data class arrives, the model refers to this base knowledge to efficiently learn the new task by leveraging what has already been learned.
>
> Specifically, the model computes task affinity to identify the most closely related source tasks. It then heavily utilizes these closest tasks to accelerate the learning of the new class. We observe that catastrophic forgetting primarily occurs in these closely related tasks, as their associated weights are updated most significantly. Therefore, a replay strategy that focuses on these closest tasks, rather than all previously learned tasks, is more efficient and effective.
>
> 8. > "... why not use a pre-trained image classification model to identify the closest classes..."
>
> Using a separate module to measure task affinity is a viable option and has been explored in several prior works. However, these approaches typically require training a new classification model and a substantial number of data points (from both the source and target data classes) to accurately compute the affinity score. In contrast, our method leverages the existing discriminator from the GAN model, eliminating the need for a separate classification model and requiring only a few data samples from the target task to efficiently and reliably measure task similarity.
>
> We are grateful for your insightful feedback and look forward to any additional suggestions you may have.

---

> > ### Comment · Reviewer_GtFb · 2025-05-25
> >
> > Thanks for the response.
> >
> > The author address the concern partially.
> >
> > Point 5 and 8 are partially addressed.
> >
> > Also, the paper explored the limited number of task sequence (six), which does not check the robustness of the model for a long task sequence.

---

> > > ### Author Response · Authors · 2025-05-27
> > > **Response to Reviewer GtFb**
> > >
> > > Thank you for your feedback. We would like to highlight the main contribution of our paper: the design of a continual GAN model capable of learning new data classes without requiring predefined labels. In contrast, existing approaches typically require manual label assignment to each data group in advance, which limits the scope of continual learning and primarily focuses on mitigating catastrophic forgetting. Our method introduces a dynamic label assignment mechanism that assigns weighted labels to new data classes as they arrive. This allows our model to handle new data without prior knowledge of the total number of classes. We have compared our method with existing continual GAN approaches, as you suggested, and demonstrated that it offers greater flexibility while maintaining strong performance on both newly introduced tasks and previously learned ones.
> > >
> > > As you suggested earlier, task similarity can also be assessed using alternative approaches. In Table 4 (Appendix), we compare our method with one that uses the FID score to measure task similarity. In the transfer learning setting, where a single new task is introduced, our approach demonstrates performance that is either comparable to or superior to the alternative, across 10-shot, 20-shot, and 100-shot learning scenarios. This suggests that our task distance metric is on par with the FID score, despite the latter requiring a large Inception-v3 model for computation. In cases where the FID score is not applicable, task similarity models (e.g., classification-based methods) demand access to substantial amounts of training data and memory cache to train and evaluate task similarity. In contrast, we propose an alternative in which the GAN generative model can directly measure the similarity between learned tasks and incoming data. Our results show that this measured distance is both meaningful and comparable to the FID score, and we further demonstrate that the distances obtained are consistent across multiple runs (Figure 2 illustrates the mean and standard deviation of task distances).
> > >
> > > Due to computational resource limitations and the size of our conditional GAN model, we conduct continual learning on a data stream with 6 data classes, which is consistent with many other approaches. However, in our case, the model has already been trained with 10 base tasks, bringing the total number of learned tasks to 16. With these 16 tasks, we demonstrate that the model is able to retain learned knowledge effectively, without suffering from catastrophic forgetting.
> > >
> > > We appreciate your constructive feedback. If you have any further questions or concerns, please let us know.

---

### Decision · Action_Editor_MVHx · 2025-06-11

**Recommendation:** Reject

**Audience:**

Yes

**Audience Explanation:**

Generative modeling is definitely of interest and continual learning is a particularly challenging problem. However, the empirical evaluation is quite limited.

**Claims And Evidence:**

No

**Claims Explanation:**

This paper presents a mode-aware GAN framework for continual generative learning, using a discriminator-based Mode Affinity Score (dMAS) to identify relevant source modes (classes) for replay and adaptation. The idea of leveraging Fisher information from the discriminator's input gradients to compute task similarity is simple and the empirical results show improvements over some existing GAN-based baselines across several datasets.

The paper received 3 reviews. While the reviewers appreciate the idea for its simplicity (basically, of the dMAS metric used by the approach, which is otherwise a simple extension of standard conditional GAN), a number of issues were raised as well.

The key point of consideration is whether the paper provides sufficient evidence for the claimed contributions. In this regard, the following points are important to highlight:

- Evaluation limited tto short task sequences: The paper claims that the method can effectively address catastrophic forgetting, a key desideratum of continual learning. It is well-known in the continual learning research community (also evidenced in prior works) that catastrophic forgetting tends to accumulate with longer sequences. For a method claimed to handle catastrophic forgetting, an evaluation that is limited to 6 tasks is not sufficient, and evaluation on longer task sequences becomes essential (most recent methods consider much longer task sequences). Continual learning becomes significantly harder as the number of tasks increases, and the similarity across tasks decreases. Without showing performance under long, heterogeneous sequences, the model may work well only in toy or short-horizon settings, which limits its practical relevance.

- Comparison with SOTA: The method is claimed to achieve comparable performance to SOTA methods but the baselines are rather old (till 2021) and aren't really considered SOTA. In particular, no comparisons are provided with non-GAN baselines such as diffusion-based continual learners or VAE variants, which are current SOTA on such problems. Given that the methodological novelty of the paper is rather slim, it needs to make a stronger case for it to be considered an alternative to such methods. Unfortunately, the paper doesn't present any such evidence.

- The proposed dMAS score is the primary contribution of the paper but it is quite similar to (and loosely inspired by) other task/mode similarity metrics such as Task2Vec and Fisher Kernel. Moreover, the dMAS score is only loosely validated -- while it is shown to be consistent, there is no evidence that it outperforms simpler alternatives like task similarities computed using pretrained embeddings. There is no ablation on such dMAS alternatives and no comparison to simpler task similarity metrics (e.g., cosine distance of discriminator features, CLIP embeddings, etc.).

After the initial round of review, the authors submitted an updated manuscript which was considered but it didn't have much to address the concerns, except an experiment with an additional GAN-based baseline StyleCL (but didn't address the concern related to evaluation in more complex and realistic CL settings).

In the end, the concerns still persisted because of the lack of strong empirical or rigorous evidence for the claimed contributions. Therefore, the paper does not meet the acceptance bar for TMLR.